# Experimental Analysis of Wax Micro-Droplet 3D Printing Based on a High-Voltage Electric Field-Driven Jet Deposition Technology

**Yanpu Chao** [1,*], **Hao Yi** [2,3,*], **Fulai Cao** [1], **Yaohui Li** [1], **Hui Cen** [1] **and Shuai Lu** [1]

1   Department of Mechanical Engineering, Xuchang University, Xuchang 461002, China; 13271247588@163.com (F.C.); lyh@xcu.edu.cn (Y.L.); hui.cen@foxmail.com (H.C.); zdlu996@126.com (S.L.)
2   State Key Laboratory of High Performance Complex Manufacturing, Central South University, Changsha 410083, China
3   State Key Laboratory of Mechanical Transmission, Chongqing University, Chongqing 400044, China
*   Correspondence: chaoyanpu@163.com (Y.C.); haoyi@cqu.edu.cn (H.Y.)

**Abstract:** High-voltage electric field-driven jet deposition technology is a novel high resolution micro scale 3D printing method. In this paper, a novel micro 3D printing method is proposed to fabricate wax micro-structures. The mechanism of the Taylor cone generation and droplet eject deposition was analyzed, and a high-voltage electric field-driven jet printing experimental system was developed based on the principle of forming. The effects of process parameters, such as pulse voltages, gas pressures, pulse width, pulse frequency, and movement velocity, on wax printing were investigated. The experimental results show that the increasing of pulse width and duration of pulse high voltage increased at the same pulse frequency, resulting in the micro-droplet diameter being increased. The deposited droplet underwent a process of spreading, shrinking, and solidifying. The local remelting and bonding were acquired between the contact surfaces of the adjacent deposited droplets. According to the experiment results, a horizontal line and a vertical micro-column were fabricated by adjusting the process parameters; their size deviation was controlled within 2%. This research shows that it is feasible to fabricate the micro-scale wax structure using high-voltage electric field-driven jet deposition technology.

**Keywords:** micro-droplet; high-voltage electric field-driven jet; Taylor cone; micro-scale wax structure





## 1. Introduction

Micro/nano 3D printing technology is a new micro/nano manufacturing method based on the principle of additive manufacturing. It can directly print and form functional products with a micro/nano characteristic structure [1]. Compared with traditional LIGA (Lithographie, Galvanoformung and Abformung) technology [2], nano-lithography [3], micro/nano embossing [4], high-speed micro-milling, micro-electrical discharge machining (EDM) [5], and other micro/nano manufacturing technologies, micro/nano 3D printing technology has the advantages of being a simple manufacturing process, being low in cost, and of allowing for high utilization of materials, wide material application, and direct formation.

At present, the micro/nano 3D printing technologies developed mainly include: direct laser writing based on two-photon polymerization [6], micro-stereophotolithography, electro-hydrodynamic jet (E-jet) printing [7], laser-induced forward transfer (LIFT) [8], focused electron beam (FEB) induced deposition [9], electrochemical deposition [10], focused-ion-beam direct writing (FIBDW) [11], and microplasma deposition [12–14]. The printing scales include micron scales, sub-micron scales, and nano scales. Micro/nano 3D printing technologies have been applied to aerospace, micro-electro-mechanical systems, biomedicine, intelligent materials, solar cells, flexible electronics, microfluidic chips, nano-catalysis, and other fields and industries, demonstrating favorable prospects for industrial application [15–18].

However, the above existing micro/nano 3D printing technologies are unable to print multiple materials simultaneously, especially as it is difficult to achieve macro/micro/nano scale 3D printing and printing of multi-material complex 3D structures. Compared with other 3D printing processes, the freeform fabrication with micro-droplet on-demand sprayed deposition has shown outstanding advantages and potential in multi-material and multi-scale structure manufacturing.

However, the traditional inkjet printing [19] process is limited by the material, the viscosity of material ink being low, and the resolution of the printed graph being generally more than 50 microns. The metal droplet on-demand jet printing process [20] can realize the ejecting and deposition forming of a variety of metal materials, but the size of the droplet is larger than the diameter of the nozzle, which does not allow for high printing accuracy and resolution [21].

Electrohydrodynamic jet (E-jet) printing [22] is a micro-droplet spray forming technology based on electro fluid dynamics (EHD), which was proposed and developed by Park and Rogers et al. in 2007 [23]. It has the advantages of good material compatibility, low cost, simple system structure, and high resolution. At present, global researchers have carried out in-depth research on electrohydrodynamic jet printing and direct writing and have made considerable breakthroughs in many aspects [24]. Huang et al. [25–27] applied the electrohydrodynamic direct writing technology to print high-resolution nanofibers for the production of stretchable flexible electronic devices on a specific material substrate. Zhang et al. [28–30] fabricated tissue engineering scaffolds by electrohydrodynamic jet printing. Han et al. [31] printed the columnar array structure with high depth ratio by utilizing the phase transformation properties of materials. Jiang et al. [32] developed an all electrospray printing process for perovskite solar cells (PSCs) in ambient air below 150 °C. García-Farrera et al. [33] used electrohydrodynamic deposition technology to print sub-100 nm ceramic piezoelectric films. Taylor et al. [34] reported a graphene oxide low-cost nano flake conductometric gas sensor, which was fabricated by atmospheric pressure electrospray printing.

Although electrohydrodynamic jet printing technology has high printing accuracy and resolution, the printing process needs to ensure that the distance between the nozzle and the substrate is limited to 3 mm. Therefore, the electrohydrodynamic jet printing technology still has many shortcomings and limitations in printing materials, nozzle materials, deposition substrates, forming height, printing stability, and other aspects, especially as it is difficult to realize the integrated printing of macro/microstructures [35].

In order to solve the existing problems of 3D printing in multi-material and multi-scale complex 3D structure manufacturing, Lan et al. proposed and established a novel micro/nano 3D printing technology using high-electric field-driven jet deposition (HEFD) [36]. Its preliminary exploration has been carried out in the application of transparent electrode, transparent electric heating, transparent electromagnetic shielding, vascular stent, flexible electronics, paper-based electronics, and many other fields [37–40]. High-voltage electric field-driven jet deposition technology is a kind of high-resolution micro/nano scale 3D printing, which is based on electrostatic induction and electrohydrodynamic jetting behavior. The molten viscoelastic liquid in the nozzle tip is stretched into a Taylor cone under an electric field force; the high-resolution micro/nano scale 3D printing can thus be achieved, which is an order of magnitude lower than the nozzle diameter. Two working modes: pulsed cone-jet mode and continuous cone-jet mode, have been adopted for implementing multi-scale 3D printing, which are suitable for various solutions and melted materials with ultrahigh viscosity, macro/microstructures, and multi-scale fabrication.

In recent years, the research on the fabrication process of microfluidic devices based on silicon polymers has attracted increasingly more attention from researchers in different fields. The process principle is mainly to fabricate micro-channel structures by using a 3D printing wax mold combined with PDMS micro-transfer technology [41]. The analytical accuracy of the microfluidic device is directly affected by the width and section the shape of microfluidic channel structure. Given that the traditional 3D printing wax mold is

fabricated by the inkjet printer, the printing accuracy is limited by the nozzle, the printing resolution is generally close to the nozzle diameter, and the width of the micro-channel after printing is generally more than 200 μm. It is difficult to fabricate micro-channel structures that are tens of microns in width.

In this study, microcrystalline wax is used as experimental material. The working mechanism of high-voltage electric field-driven jet deposition technology was analyzed. The ejection and deposition process of the pulsed cone-jet mode was studied. Based on this, the densified wax line and the large ratio of height to diameter micro-cylinder were printed. This research proposes a new technical approach for a fabricated micro-scale wax part with tens of microns and has important application prospects in the fabrication of micro-channel structures using a 3D printing wax mold combined with PDMS micro-transfer technology.

## 2. Process Principle

High-voltage electric field-driven jet deposition technology is a novel high-resolution micro/nano scale 3D printing, which is based on electrostatic induction and electrohydrodynamic jetting behavior. Figure 1 shows a schematic diagram of the principle of on-demand high-voltage electric field-driven jet deposition.

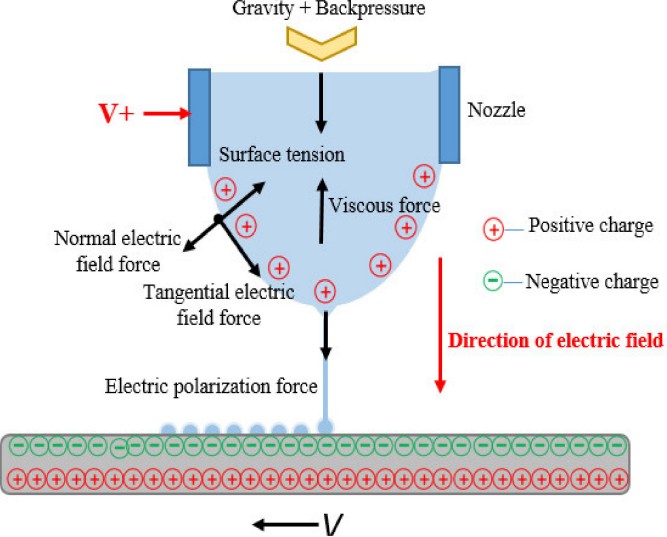

**Figure 1.** Schematic diagram of the principle of high-voltage electric field-driven jet deposition on demand.

The forming principle is as follows: the wax liquid at the bottom of the nozzle is formed into a meniscus shape under the action of gravity and gas backpressure, the conductive nozzle is connected to the positive pole of the high-voltage pulse power supply, and a high voltage is applied to the conductive nozzle to give it high potential. The wax liquid material inside the nozzle is polarized under the high voltage, and the positive charges are accumulated on meniscus shape liquid surface. When the high voltage nozzle is close to the deposition substrate, the electrostatic induction is generated between the nozzle and the substrate, resulting in the charge being migrated on the surface and inside the substrate. The charge is then redistributed on the substrate. The negative charge is distributed on the upper surface of the substrate, and the positive charge is distributed on the lower surface of the substrate. The interaction is generated between the positive charge on the meniscus shape liquid and the negative charge on the upper surface of the substrate, resulting in the electric field being formed between the meniscus liquid surface and the substrate. The meniscus liquid surface is stretched and deformed gradually into a Taylor cone under the action of electric field force, surface tension, viscosity force, and electric polarization force. As the high voltage increases, the tip charges continually accumulate. Once the electric field force exceeds the surface tension and viscosity force of the liquid, a very fine Taylor

cone jet is generated. The diameter size of jet is usually 1~2 orders of magnitude lower than the nozzle diameter. Finally, under the action of liquid surface tension, the micro-jet is broken into individual droplets and deposited on the substrate according to the frequency of a high-voltage pulse.

## 3. Experiment System

Based on the abovementioned principles, an experimental system was developed herein. The experimental system mainly included a high voltage power supply (DMC-200, Dalian Dingtong Technology China, Dalian, China), an electrode ring, an air pump (BLT-5A/7A, Bolaite, Shanghai, China), an air pressure control unit (IR2000, King Kong Pneumatic Technology, Wenzhou, China), an *XY*-axis movement platform (G3TH764, KingWonda, Shenzhen, China), a Z-axis movement platform, a deposition substrate, an annular heating furnace (customized), a quartz crucible (customized), a print head (stainless steel precision musashi needle), a nozzle heating block, and a temperature controller (OHR-A300, HongYun instrument, Shunchang, China).

Figure 2 shows a schematic diagram of a high-voltage electric field-driven jet printing experimental system. The positive pole of the high-voltage power supply was connected with the electrode ring. The electrode ring was fixed on the outer surface of the print head. The high voltage power supply can realize two power supply modes: high-voltage pulse and high-voltage DC. The annular heating furnace was fixed on the Z-axis moving platform through a connecting frame, and the quartz crucible was installed inside the annular heating furnace. The printing material was placed inside the quartz crucible, and the temperature of the annular heating furnace could be controlled through the temperature controller to achieve the melting of the printing material. The nozzle heating block was installed on the outer surface of the printing head. The nozzle temperature could be precisely controlled to meet the requirement of the injection temperature of the printing material by the temperature controller. The control accuracy of the temperature is ±0.5 °C. The deposited substrate was installed on the *XY*-axis moving platform, and the droplet material was precisely deposited on the substrate by coordination controlling the movement of the *XY*-axis moving platform and controlling the ejection frequency of the printing material. The repeated positioning accuracy of the *XY*-axis moving platform is ±0.01 mm. After one layer was printed, the printing head moved upward by a thickness layer, and a stable electrostatic induced electric field was generated between the conductive nozzle and the printed layer; the next layer was then printed. The above process was repeated until the manufacturing of the entire structure was completed.

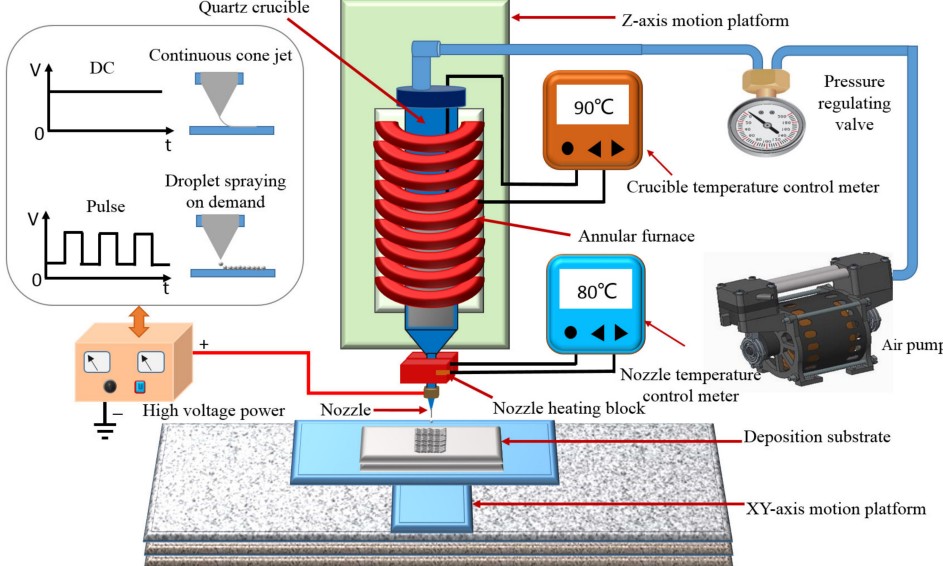

**Figure 2.** Schematic diagram of a high-voltage electric field-driven jet printing experimental system.

The injection and deposition process of micro-droplets is a high-speed nonlinear dynamic process using high-voltage electric field-driven jet deposition technology. In order to monitor the real dynamic process of Taylor cone ejection and droplet deposition in real time, a high-speed monitoring system was designed and developed, which can guide the experimental process. Figure 3 shows the working principle system. The system is mainly composed of a high-performance computer, high-speed image capture card (DH-CG410, Daheng, Beijing, China), high-speed CCD camera (Sp-5000M-CXP2, Daheng, Beijing, China), integrated microscope (RetroZoom65, Optem, Medina, OH, USA), stroboscope (DA/DB, Monado, St. Louis, MO, USA), synchronous trigger, and so on. The high-speed CCD uses ID's Os4 camera, with a resolution of 1024 × 1024 and a frame rate of up to 3000 fps, making it easy to capture images of high-speed moving objects. The image acquisition of the Silicon Software Company in Germany can transfer the collected images to the computer hard disk through PCI bus and save them for analysis and calculation. A stroboscope is used as the light source. In the experiment, the simultaneous exposure method of stroboscope and CCD camera was adopted to realize the shooting of the droplet ejecting and deposition instant image. The high-speed CCD camera and the stroboscope light source were simultaneously driven by the TTL signal provided by an external trigger to realize synchronous operation.

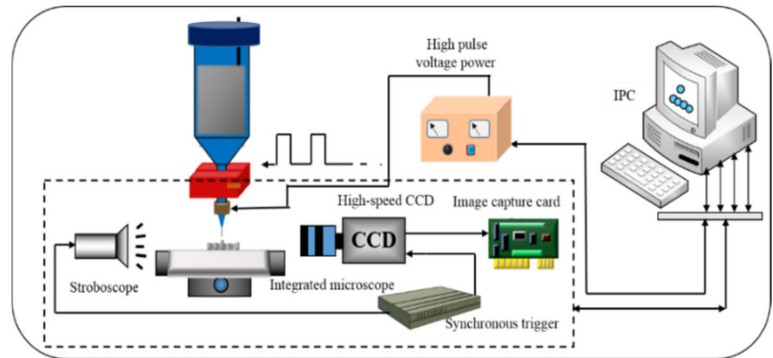

**Figure 3.** Schematic diagram of a high-speed monitoring system.

## 4. Experiment Results and Discussion

### 4.1. Arc Discharge

In order to verify the ability of accurately controlling the high-voltage electric field, an arc discharge experiment of the high-voltage electric field was carried out first in a 3D printing experiment system. A stainless-steel precision Musashi needle was selected for the nozzle, and no printing materials were in the nozzle. The conductive nozzle was connected with the positive pole of the high-voltage pulse power supply, and a high voltage was applied to the conductive nozzle to create a high potential. A copper clad plate was used for the deposition substrate without connecting the ground electrode. The frequency of the high voltage pulse was set as 1 Hz, and the voltage of the high voltage pulse was gradually changed to 1000 V, 1500 V, 2000 V, 2500 V, 3000 V, 3500 V, 4000 V and 5000 V. The arc discharge phenomenon was observed by the high-speed monitoring system.

In the experiment, when the nozzle gradually approached the deposition substrate, the electric field between the nozzle and the substrate was enhanced. Under the action of the discharge energy, the air medium between the poles was ionized and broken down, forming an arc discharge. Figure 4 shows the pictures of arc discharge obtained by a high-speed camera system under different pulse voltages. The measured lengths of discharge arc were 0 mm, 0.165 mm, 0.395 mm, 0.580 mm, 0.860 mm, 1.085 mm, 1.260 mm and 1.755 mm, respectively. The experimental results show that the phenomenon of arc discharge becomes more intense with the increase of pulse voltage. When the voltage was 1000 V, the phenomenon of arc discharge did not occur, indicating that the voltage was too low.

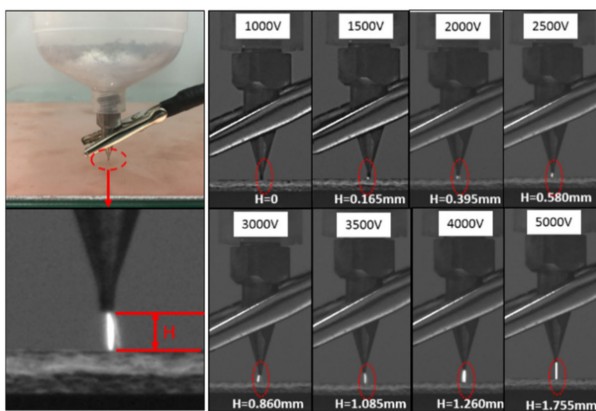

**Figure 4.** Arc discharge phenomenon under different pulse voltages.

### 4.2. Liquid Morphology at the Bottom of the Nozzle

In the process of micro-droplet ejecting and depositing driven by a high electric field, it is necessary to ensure that the liquid surface at the bottom of the nozzle is a meniscus shape. On this basis, a Taylor cone can be generated under the action of a high-voltage electric field, and micro-droplet on-demand ejecting can be achieved. Microcrystalline wax 80$^{\#}$ (CAS NO: 63231-60-7) was used as the printing material in the experiment, which can be obtained by solvent fractionation of the still-bottom fraction of petroleum by suitable dewaxing or deoiling. Microcrystalline wax is generally composed of fine needle-like or granular crystals. The main components are cycloalkanes and aromatic hydrocarbons with larger molecular weight and longer carbon chains. Its molecular formula is $C_{11}H_{14}N_2$, and its molecular weight is 174.24226 g/mol. Its melting point is $T_m$ is 80 °C, the glass transition temperature is $T_g$ is 67.5 °C, the dynamic viscosity is 16.5 cP, and the surface tension is about 0.245 N·m$^{-1}$ after being melted into a liquid state at 80 °C. Under the action of gravity, the wax liquid will naturally drop from the nozzle or form large wrapped droplets at the nozzle. Therefore, it is necessary to utilize a technical approach to make the wax liquid form a meniscus shape at the nozzle and maintain a stable meniscus shape. At present, there are two commonly used methods: (1) applying negative pressure to balance the wax liquid's own gravity and (2) coating the nozzle with a hydrophobic film. In this paper, the negative pressure balance gravity method is used to make the wax liquid form meniscus shape at the nozzle and maintain a stable meniscus shape.

Figure 5 shows the morphologies of wax liquid at the bottom of the nozzle under different gas pressures. When the negative pressure is set as 0 Pa, the bottom of the nozzle is completely wrapped by wax liquid, which affects the printing stability and accuracy as shown in Figure 5a. As shown in Figure 5b, when the negative pressure is set as 450 Pa, the larger wrapped droplet at the bottom of the nozzle essentially disappear; a stable meniscus shape liquid is generated. When the negative pressure increases to −750 Pa, the meniscus shape liquid disappears due to excessive negative pressure as shown in Figure 5c. The Taylor cone cannot be generated, and micro-droplet on-demand ejecting cannot be achieved under the action of a high voltage electric field. Therefore, the optimal meniscus shape liquid can be guaranteed under −450 Pa negative pressure, and too high or too low negative pressure is not conducive to the droplet spray forming.

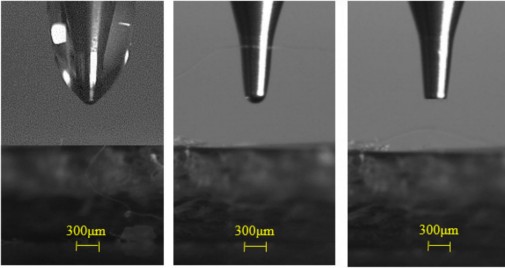

**Figure 5.** Morphology of wax liquid at the bottom of the nozzle under different gas pressures: (**a**) fully wrapped nozzle; (**b**) meniscus shape; (**c**) no meniscus shape.

### 4.3. Taylor Cone and Droplet Ejecting

Two working modes, pulsed cone-jet mode and continuous cone-jet mode, are adopted in high voltage electric field driven jets, according to different printing materials and printing characteristic structures. The pulsed cone-jet mode uses a high-voltage pulse as a driving signal, and the continuous cone-jet mode uses a high-voltage DC as a driving signal.

The high-voltage pulse is a digital signal, the pulse frequency, pulse width, and duty cycle can be adjusted in real time, which makes it more robust as there are numerous possibilities. When using a high-voltage pulse, the printing discretized into voxels. For low viscosity materials or fine structures, the pulsed cone-jet printing mode is used to ensure the accuracy of the printed micro-characteristic structures. The high-voltage DC is a continuous analog signal, where the electric field is continuous. The printed material is deposited in a continuous jet stream and will not be separated into droplets. For high-viscosity materials or macroscopic structures, the continuous cone-jet mode is adopted to achieve efficient printing under the premise of meeting the required accuracy.

In this paper, the pulse cone-jet mode was studied. Under the action of high-voltage pulse, the wax liquid material at the bottom of the conductive nozzle went through the following processes: from tensile deformation to Taylor cone to jet retraction and finally to fracture into a single droplet. The diameter of the droplet produced by the Taylor cone jet is much smaller than the diameter of the nozzle, this is achieved using precise control of the pulse width and the pulse frequency.

The experimental process parameters are presented in Table 1. The microcrystalline wax was selected as the printing material, and the stainless-steel precision Musashi needle with inner diameter 200 μm and outer diameter 350 μm was selected for the nozzle. The negative pressure in the crucible was set as 450 Pa, the temperature of the crucible $T_c$ was set as 90 °C, and the temperature of the nozzle $T_n$ was set as 80 °C. The Pulse frequency $f$ was set as 1 Hz, the pulse width $W_t$ was set as 10 ms, and the applied pulse voltage $U$ was set as 1500 V. The experiment process was observed with a high-speed camera. The arc discharge phenomenon was observed by the high-speed monitoring system.

**Table 1.** Process parameters of Taylor cone and droplet ejecting.

| Printing Material: | Microcrystalline Wax |
|---|---|
| Back pressures of crucible: $P$ (Pa) | −450 |
| Diameter of the nozzle: $D$ (μm) | 200 |
| Temperature of the crucible: $T_c$ (°C) | 90 |
| Temperature of the nozzle: $T_n$ (°C) | 80 |
| Pulse frequency: $f$ (Hz) | 1 |
| Pulse width: $W_t$ (ms) | 10 |
| Pulse voltage: $U$ (V) | 1500 |

Figure 6 shows the whole process from Taylor cone jet generation to droplet deposition in the high-voltage electric field-driven jet printing experiment photographed by a high-speed camera system. At 0 ms, because the high-voltage pulsed electric field was not applied, the liquid meniscus at the bottom of the nozzle was not particularly obvious, and the height of the liquid meniscus was lower. At 5 ms, the meniscus liquid at the bottom of the nozzle began to be stretched and deformed, the height of the meniscus liquid was increased, and the meniscus liquid appeared as a cone under the action of a high-voltage electric field. At 6–7 ms, the meniscus liquid at the bottom of the nozzle was further stretched into a Taylor cone jet. At 8–9 ms, the Taylor cone jet at the bottom of the nozzle began to retract and fracture into a single droplet deposited on the substrate. At 11 ms, the Taylor cone jet retraction fracture was completed, and the meniscus liquid surface at the bottom of the nozzle was restored to the minimum height. The single droplet electric field driven injecting was completed and prepared for the next injecting. The morphology of the micro-droplet deposited on the substrate was observed; the diameter of the single micro-droplet after deposition and solidification was about 42 μm.

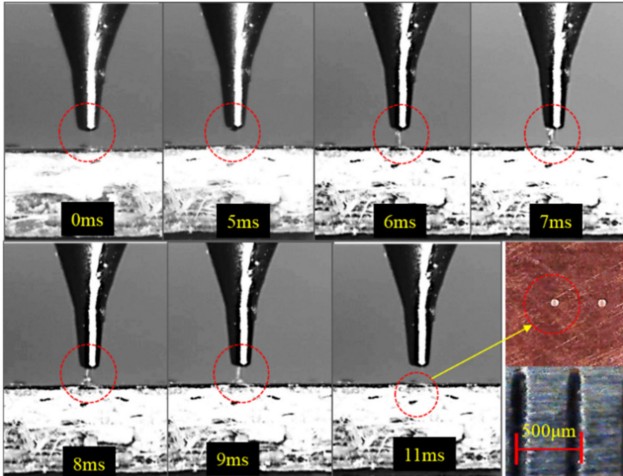

**Figure 6.** Image of Taylor cone generation and droplet eject deposition process.

*4.4. Pulse Width*

By adjusting the frequency and pulse width of the high-voltage pulse signal, the desired duration and interval time of the high-voltage pulse can be obtained, and then the droplet size can be controlled. In the experiment, the duration of the high-voltage pulse was changed by adjusting the pulse width when the pulse frequency remained constant, and the effect of the process parameters on the forming size of the droplet was investigated.

The microcrystalline wax was selected as printing material, and the copper clad plate was used as a deposition substrate. The inner diameter of the nozzle was 150 μm, and the outer diameter of the nozzle was 300 μm. The same experimental process parameters, as shown in Table 1, were employed except for the pulse width $W_t$. Figure 7 shows the experimental results of droplets deposition with different pulse widths. The pulse width was set and adjusted by a precision function generator (AFG2000-SC, TektroniX, Shanghai, China), the parameter error was ±0.1 ms. The pulse widths $W_t$ is set as 10 ms, 20 ms, 30 ms, 50 ms, 100 ms, 300 ms and 500 ms in the experiment. The diameter of deposited droplets under different pulse widths were measured by Image Pro software, with the measurement error set as ±0.01 μm. Each measurement droplet was measured three times and the average values were calculated as shown in Figure 7. The average diameters of the measured droplets were 37.69 μm, 43.98 μm, 48.75 μm, 56.54 μm, 60.58 μm, 69.11 μm and 75.39 μm, respectively. The droplet diameter versus pulse width is an approximate linear relationship in different ranges. Figure 8 shows the linear relationship plot.

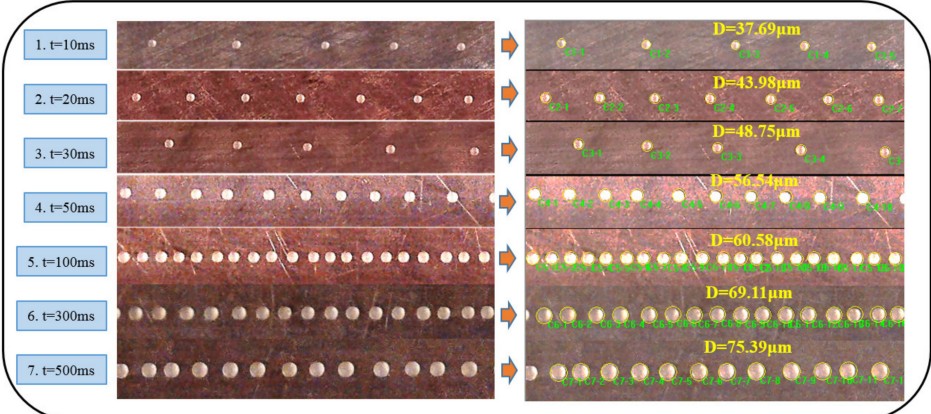

**Figure 7.** The experimental results of the droplet deposition at different pulse widths.

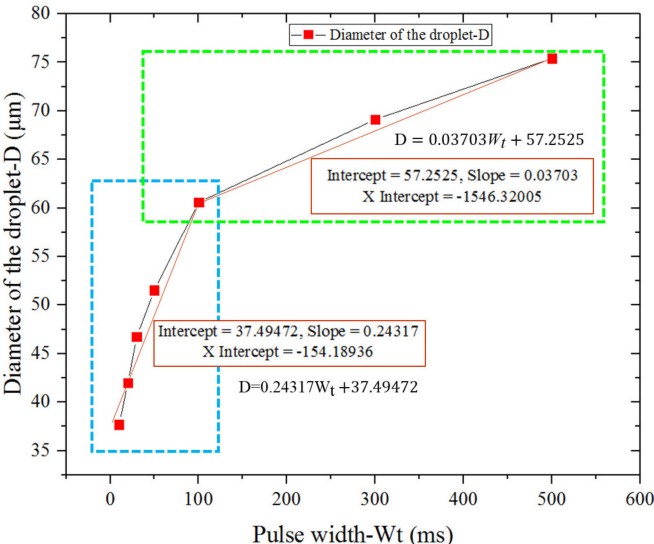

**Figure 8.** The approximate linear relation of droplet diameter versus pulse width in different ranges.

Theoretically, a complete droplet jetting process needs a high voltage to last for a certain period of time, which mainly includes electric charge accumulation release and deformation fracture time of the Taylor cone jet. If the duration of the high-voltage pulse is too short, the deformation fracture of the Taylor cone jet cannot be completed, and the droplet jet cannot be realized. With the increasing of pulse width $W_t$, the duration of the pulse high voltage was increased at the same pulse frequency, and the amount of liquid ejected from the Taylor cone was also aggrandized, resulting in the micro-droplet diameter being increased.

### 4.5. Horizontal Direction Deposition

In the experiment, the movement velocity of the deposition platform $V_p$ was changed to observe its influence on the deposition distance of droplets. The experimental process parameters are shown in Table 2. Different movement velocities of the deposition platform $V_P$ were set (the values are listed in Table 2). Figure 9 shows the images of the droplet printing and deposition process at different velocities in a horizontal direction obtained by a high-speed camera system. The morphology of the deposited lines at each movement velocity was captured using a macro lens (shown in Figure 10).

**Table 2.** Process parameters of droplet deposited lines.

| Printing Material | | Printing Substrate | | Diameter of the Nozzle: $D$ (μm) | | Pulse Frequency: $f$ (Hz) | |
|---|---|---|---|---|---|---|---|
| Microcrystalline wax | | PET (Polyethylene terephthalate) | | 200 | | 1 | |
| Pulse width: $W_t$ (ms) | | Temperature of the crucible: $T_c$ (°C) | | Temperature of the nozzle: $T_n$ (°C) | | Pulse voltage: $U$ (V) | |
| 10 | | 90 | | 80 | | 1500 | |
| Movement velocity of the deposition platform: $V_p$ (Plus/s) | | | | | | | |
| 1 | 2 | 3 | 4 | 5 | 6 | 7 | 8 |
| F100 | F90 | F70 | F50 | F30 | F10 | F5 | F2 |

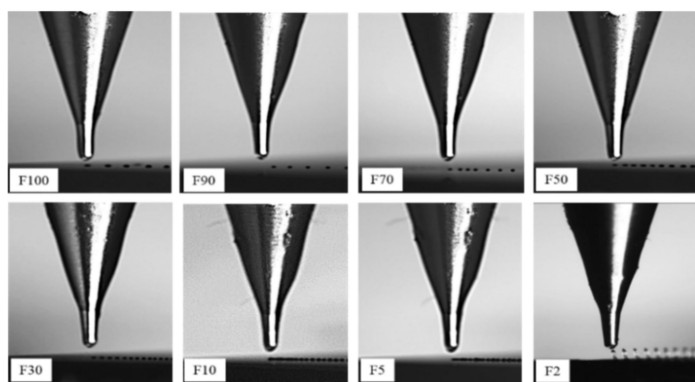

**Figure 9.** Image of droplets deposition process at different velocities in horizontal direction.

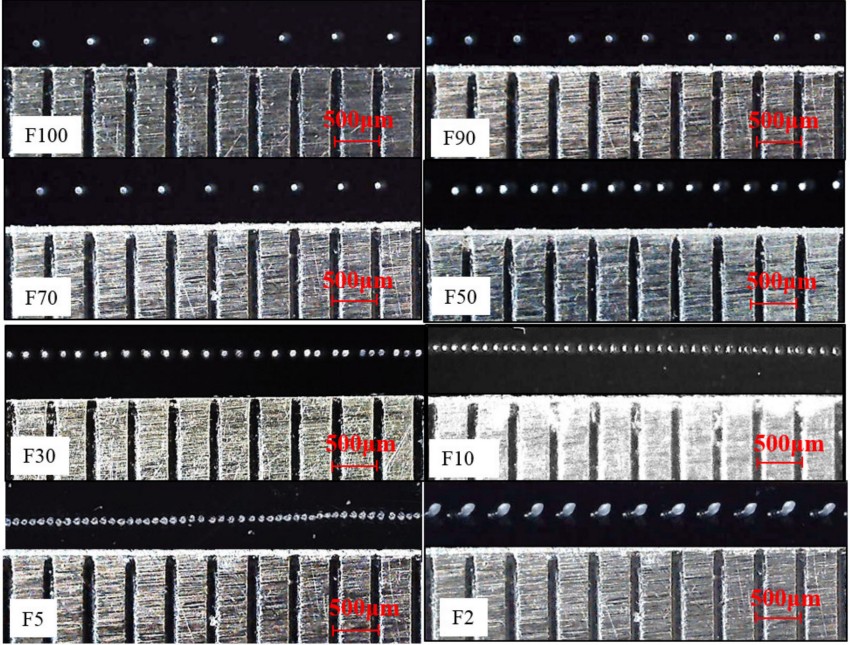

**Figure 10.** The morphology of the deposited lines at different movement velocities.

The movement velocity of the deposition platform was set and adjusted by a precision motion controller (y, GUC, GooGoLTECH, Shenzhen, China), with a parameter error of ±1 Plus/s. The movement velocities of the deposition platform $V_P$ were set as F100, F90, F70, F50, F30, F10, F5 and F2 in the experiment. The spacing of the deposited droplets under different movement velocities were measured by Image Pro software, and the measurement error was set as ±1 μm. Each measurement object was measured three times and the average values are calculated. The average measured spacing of the deposited droplets S

were 668 µm, 580 µm, 475 µm, 332 µm, 234 µm, 124 µm, 85 µm and 34 µm, respectively. The droplet spacing versus speed of the deposition platform is an approximate linear relationship; Figure 11 shows the linear relationship plot.

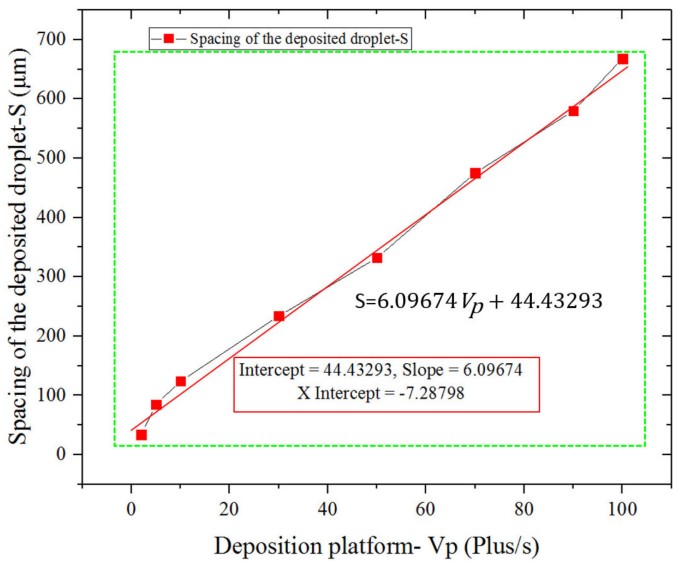

**Figure 11.** The approximate linear relation of droplet spacing versus speed of the deposition platform.

It can be seen from the experimental results that, with the decreasing of the movement velocities ($V_p$), the distance between droplets after deposition was also gradually decreased. When the movement velocitiy, $V_p$, was set as F5, the deposited droplets form a continuous line. However, there were still small gaps between adjacent droplets, the main reason being that the average droplet spacing of 85 µm was larger than the average droplet diameter of 72 µm after deposition. When the movement velocity, $V_p$, was set as F2, the deposited droplets formed an array of uniformly spaced micro-cylinders, the main reason being that the average droplet spacing of 34 µm was less than the average droplet diameter of 72 µm after deposition. The overlap rate of two adjacent droplets was too high, which led to the deposited droplets growing upwards and being unable to form a continuous line. The diameter of the deposited droplets did not change very much under different movement velocity, $V_p$, the average diameter of the deposited droplets was determined to be 72 µm using software measurement. These experimental results indicate that the movement velocities ($V_p$) have almost no effect on the diameter of droplet after deposition.

### 4.6. Vertical Direction Deposition

Figure 12 shows the image of the droplets vertical deposition process in the vertical direction obtained by a high-speed camera system. In the experiment, the deposition substrate remained stationary, and the Z-axis movement velocity was set as F100. Other experimental process parameters are shown in Table 2. Figure 12a–d show the deposition process of the first, second, third, and fourth droplet, respectively. It can be seen that the jetting process of each wax droplet has undergone four processes: tensile deformation to Taylor cone to jet retraction then to fracture into a single droplet.

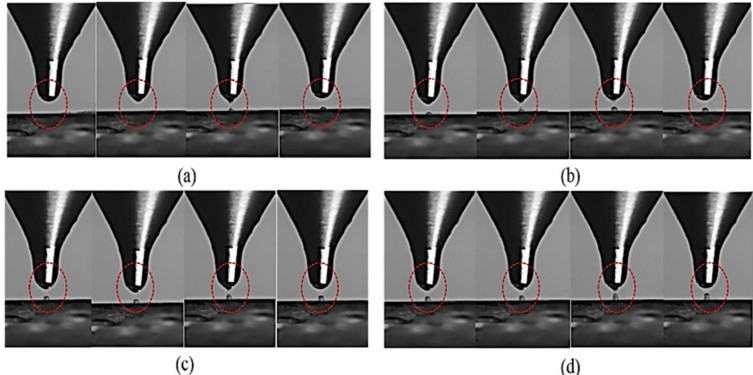

**Figure 12.** A vertical deposition process of droplets on demand: (**a**) the first droplet deposited; (**b**) the second droplet deposited; (**c**) the third droplet deposited; (**d**) the fourth droplet deposited.

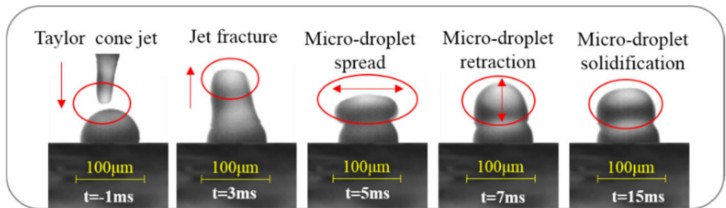

**Figure 13.** The process of droplet spreading, shrinking, and solidifying in vertical deposition.

### 4.7. Deposition Line and Column

In order to further verify the feasibility of micro/nano 3D printing based on high-voltage electric field-driven jet deposition, on the basis of the above experimental research and analysis, the horizontal line and vertical column were fabricated using wax micro-droplets.

The microcrystalline wax was selected as printing material, and the copper clad plate was used as a deposition substrate. The inner diameter of nozzle $D$ was 200 μm, the negative pressure in the crucible was set as 450 Pa, the temperature of the crucible $T_c$ was set as 90 °C, and the temperature of the nozzle $T_n$ was set as 80 °C. The pulse frequency $f$ was set as 10 Hz, and the applied pulse voltage $U$ was set as 1600 V.

Figure 14 shows the deposited lines of wax droplets on a copper clad plate, the deposited line with one layer is shown in Figure 14a, and the deposited line with ten layers is shown in Figure 14b. It can be seen that there is no obvious gap between the adjacent deposited droplets, and a good remelting and bonding state of droplets has been obtained. The formed wax lines have a compact structure and good straightness. Four positions on each deposited line were selected as line width measurement points. According to the software measurement analysis, the line widths of the deposited line with one layer were L1 = 87.65 μm, L2 = 88.63 μm, L3 = 84.68 μm and L4 = 86.68 μm in the four positions; the line widths of the deposited line with ten layers were L5 = 85.64 μm, L6 = 87.35 μm, L7 = 86.35 μm and L8 = 85.86 μm in the four positions. According to the analysis and measurement results, the line widths of the deposited lines were essentially the same, and the size deviation was controlled within 2%.

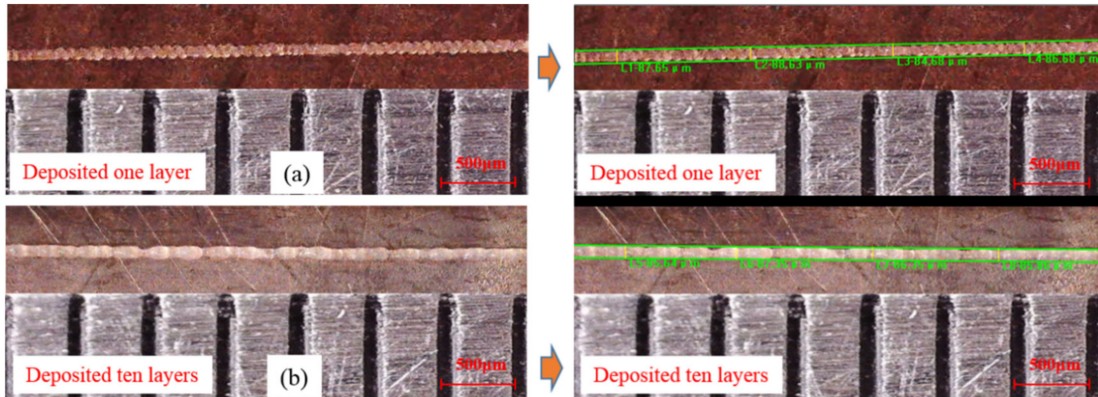

**Figure 14.** The deposited lines by wax droplets on a copper clad plate: (**a**) deposited one layer; (**b**) deposited ten layers.

Figure 15 shows the deposited micro-cylinder of wax droplets in a vertical direction with a large ratio of height to diameter. It can be seen that the local remelting was acquired between the contact surfaces of deposited droplets, and a relatively strong bonding was formed. The formed wax micro-cylinder with a large ratio of height to diameter has a compact structure without pores, a smooth surface, and good verticality. Four positions (the location of the refusion region of two adjacent droplets) on the deposited micro-cylinder were selected as diameter measurement points. According to the software measurement analysis, the diameter of the deposited micro-cylinders were $D1 = 80.34$ μm, $D2 = 83.65$ μm, $D3 = 82.86$ μm and $D4 = 81.32$ μm in the four positions. According to the analysis and measurement results, the diameter of the deposited micro-cylinders were essentially the same, and the size deviation was controlled within 2%.

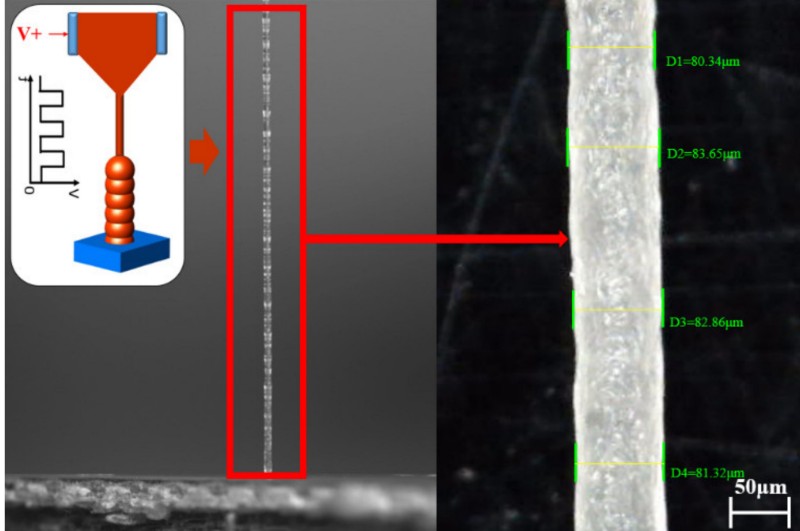

**Figure 15.** The deposited micro-cylinder of wax droplets with large ratio of height to diameter.

In conclusion, the deposition experiment results of the above line and micro-column verify the feasibility of micro/nano 3D printing based on high-voltage electric field-driven jet deposition. This research proposes a new technical approach for a fabricated micro scale wax structure with tens of microns and has important application prospects for the fabrication of micro-channel structures by using a 3D printing wax mold combined with PDMS micro-transfer technology.

In order to improve the spraying efficiency of micro-droplets driven by a high-voltage-electric field, a kind of parallelization, multiplexed, and array injection device will be

developed. It could increase the magnitude of the throughput by two or more orders without sacrificing the quality of the print, greatly extending the capabilities of the technology.

### 5. Conclusions

The conclusions drawn from this paper are as follows:

(1)    A high-voltage electric field-driven jet deposition technology was proposed to fabricate wax micro-structures. The mechanism of the Taylor cone generation and droplet eject deposition were analyzed, and an experimental system was developed.

(2)    The droplet on-demand ejecting of a pulse cone-jet mode was studied under the action of a high-voltage pulse, the production of a droplet required the following four processes: tensile deformation to Taylor cone to jet retraction and finally to fracture into a single droplet. The diameter of the droplet was much smaller than the diameter of the nozzle.

(3)    A meniscus shape liquid at the bottom of the nozzle is necessary to generate the Taylor cone jet; the optimal meniscus shape wax liquid can be guaranteed under 450 Pa negative pressure. With the increasing pulse width $W_t$, the duration of the pulse high-voltage plus was increased at the same pulse frequency, resulting in the micro-droplet diameter being increased. Under the combined action of inertia force and gravity, the deposited droplet underwent a process of spreading, shrinking and solidifying. Local remelting and bonding were acquired between the contact surfaces of adjacent deposited droplets.

(4)    The horizontal line and vertical micro-column were fabricated using wax micro-droplets, and the size deviation between them was controlled within 2%. The correctness and feasibility of micro 3D printing based on high-voltage electric field-driven jet deposition were verified.

**Author Contributions:** Y.C. and H.Y. conceived the idea, developed the theory, performed the experiments and parameter optimization, and analyzed the results. Y.C. wrote the paper, and H.Y. provided corrections. H.C. and S.L. helped to evaluate the idea and engaged in discussions regarding the outcome. Y.L. and F.C. assisted with the editing of the paper. All authors have read and agreed to the published version of the manuscript.

**Funding:** This work was financially supported by the National Natural Science Foundation of China (grant nos. 51305128 and 52005059), Open Research Fund of State Key Laboratory of High-Performance Complex Manufacturing, Central South University (Kfkt2020-10), the key scientific research projects of the colleges and universities of Henan province (grant no.18A4600050), and the Outstanding Young Backbone Teachers projects of Xuchang University.

**Data Availability Statement:** All the data reported here can be made available on request.

**Conflicts of Interest:** The authors declare no conflict of interest.

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
