# Peer review of "Experimental Analysis of Wax Micro-Droplet 3D Printing Based on a High-Voltage Electric Field-Driven Jet Deposition Technology"

_crystals, doi:10.3390/cryst12020277_

Round 1

Reviewer 1 Report

Authors reported the wax printing process in micro-scale using a high-voltage-electric-field-driven jet deposition. Although the paper addressed a sound manufacturing engineering, I feel that it is weak from materials science point of view.

Comments:

Viscoelastic properties depend on the molecular weight of samples. Hence, authors should mention the molecular weight of wax sample and its chemical structure. This wax has Tg? (Authors mentioned only Tm). What about polarity of this wax compared to the substrate (PET or copper clad plate) because wax-substrate interactions are affected by these polarities. In some figures, if authors input a scale bar, it should be better for understanding the size of set-up.

Line 215:   16.5 mPa·s => Please convert to cP (centipoise) and compare with water. (I think this molten wax’s viscosity is comparable with that of water) (Also, the molecular weight of wax is very small)

Line 37, 61, 91: “home and abroad” is mentioned by three times (You may consider a different expression)

Line 84:  molten liquid => molten viscoelastic liquid

Line 100: microcrystalline => Author should check the crystallite size of your wax samples (using XRD).

Line 112: In Figure 1, authors input some ion symbols in a liquid, which I looks weird to me. (The wax may not contain any ionic species except for some impurities) (Maybe, the cation symbols is for nozzle, but authors input these cations into a molten wax liquid). Please check it.

Line 122: electrostatic interaction between nozzle and substrate… But your figure, there are charges in the wax liquid. (If there are charges in a liquid, there should be coulombic interaction, which may bring a different result under applied voltage)

Line 132-133: surface tension: how much in your samples?

                      viscosity force: roughly, how much?

“Once the electric field force exceeds the surface tension and viscosity force of the 132 liquid, a very fine Taylor cone jet is generated”

Line 157: check English grammar (e.g.  controller, The deposition ~~~)

Line 214:  melting point is 80 C .

                Then, what about Tg (glass transition temperature)???

Line 215:  dynamic viscosity => Please compare with water’s viscosity 

Line 221: Figure 5 (maybe, scale bar is needed. Check all your figures)

Line 248: microcrystalline ???? => Crystallite size of your sample???? And morphology of a deposited wax film????

Line 256: Physical parameter should be italic

Line 262:  0ms => 0 ms  (Gap is needed. Check all)

Line 282:    the copper clad plate was used as a deposition substrate.  

                 (Polarity of this surface and interaction with your wax???)

Line 316:   PET   => Full name may be needed for introduction.

                      Difference between the copper-clad plate and PET???
                      All physical parameter should be italic

Table 2: Printing substrate: only PET?  Or copper-clad plate together?

Line 406: Wax microstructure?

               Authors did not show any microstructure (e.g., nano/micro-scale morphology of a solidified wax film). Simply, authors may think that micro-scale deposit of wax solid on a substrate is a wax's microstructure. However, in Materials Science, when we say “microstructure, it has a different meaning. 

Author Response

Dear Editors and Reviewers:

Thank you for your letter and for the reviewers’ comments concerning our manuscript entitled “Experimental Analysis of Wax Micro-Droplets 3D Printing Based on High-voltage-Electric-Field- Driven Jet Deposition Technology” (crystals-1595188). Those comments are very valuable for improving our paper. We have studied comments carefully and have made correction. Revised portion are marked in red in the paper. The revised details according to the reviewer's comments as follows:

>Reviewer #1:

Comments and Suggestions for Authors:

Authors reported the wax printing process in micro-scale using a high-voltage-electric-field-driven jet deposition. Although the paper addressed a sound manufacturing engineering, I feel that it is weak from materials science point of view.

Comments: Viscoelastic properties depend on the molecular weight of samples. Hence, authors should mention the molecular weight of wax sample and its chemical structure. This wax has Tg? (Authors mentioned only Tm). What about polarity of this wax compared to the substrate (PET or copper clad plate) because wax-substrate interactions are affected by these polarities. In some figures, if authors input a scale bar, it should be better for understanding the size of set-up.

  1. Line 215: 16.5 mPa·s => Please convert to cP (centipoise) and compare with water. (I think this molten waxs viscosity is comparable with that of water) (Also, the molecular weight of wax is very small)

Answer: 16.5mPa·s=16.5cPs, the dynamic viscosity of water is 1cPs (20℃,101.325kPa). Microcrystalline wax 80# (CAS NO: 63231-60-7) is used as the printing material in the experiment, which is obtained by solvent fractionation of the still-bottom fraction of petroleum by suitable dewaxing or deoiling. Microcrystalline wax is generally composed of fine needle-like or granular crystals. The main components are cycloalkanes and aromatic hydrocarbons with larger molecular weight and longer carbon chains. Its molecular formula is C11H14N2, Its molecular weight is174.24226g/mol. The above contents have been added to the article.

  1. Line 37, 61, 91: “home and abroad” is mentioned by three times (You may consider a different expression)

Answer: The expressions of “at home and abroad” have been revised according to the reviewer's suggestion.

  1. Line 84: molten liquid => molten viscoelastic liquid

Answer: The molten liquid has been revised to molten viscoelastic liquid according to the reviewer's suggestion in this paper.

  1. Line 100: microcrystalline => Author should check the crystallite size of your wax samples (using XRD).

Answer: The crystallite size of microcrystalline wax 80#is less than 55 nm based on references.

  1. Line 112: In Figure 1, authors input some ion symbols in a liquid, which I looks weird to me. (The wax may not contain any ionic species except for some impurities) (Maybe, the cation symbols is for nozzle, but authors input these cations into a molten wax liquid). Please check it.

Answer: The cation symbols stand for the Positive charge on the surface of liquid, which has been illustrated and explained in Figure 1.

  1. Line 122: electrostatic interaction between nozzle and substrate… But your figure, there are charges in the wax liquid. (If there are charges in a liquid, there should be coulombic interaction, which may bring a different result under applied voltage)

Answer: First, the conductive nozzle is connected to the positive pole of the high-voltage pulse power supply to make it have a high potential. The wax liquid material inside the nozzle is polarized under the high voltage, the positive charges are accumulated on meniscus shape liquid surface. Second, when the high voltage nozzle is close to the deposition substrate, the electrostatic induction is generated between the nozzle and the substrate. The negative charge is distributed on the upper surface of the substrate, the positive charge is distributed on the lower surface of the substrate. And then, the interaction is generated between the positive charge on the meniscus shape liquid and the negative charge on the upper surface of the substrate, resulting in the electric field is formed between the meniscus liquid surface and the substrate. Finally, the meniscus liquid surface is stretched and deformed gradually into a Taylor cone under the action of electric field force, surface tension, viscosity force and electric polarization force.

  1. Line 132-133: surface tension: how much in your samples? Viscosity force: roughly, how much? Once the electric field force exceeds the surface tension and viscosity force of the 132 liquid, a very fine Taylor cone jet is generated”

Answer: The surface tension of microcrystalline wax 80# is about 0.245 N.m-1 at 80℃, and the viscosity force of microcrystalline wax 80# is about 16.5mPa·s at 80℃Provided by the manufacturer.

  1. Line 157: check English grammar (e.g. controller, The deposition )

Answer: The incorrect grammar has been corrected in this section according to the reviewer's suggestion.

  1. Line 214: melting point is 80 C .Then, what about Tg (glass transition temperature)?

Answer: The glass transition temperature Tg is 67.5℃Provided by the manufacturer.

  1. Line 215: dynamic viscosity => Please compare with water’s viscosity

Answer: The dynamic viscosity of water is 1cPs (20℃,101.325kPa), the dynamic viscosity of microcrystalline wax 80# is16.5mPa·s(20℃,101.325kPa).

  1. Line 221: Figure 5 (maybe, scale bar is needed. Check all your figures)

Answer: The scale bar has been added to Figure 5 according to the reviewer's suggestion.

  1. Line 248: microcrystalline ? => Crystallite size of your sample? And morphology of a deposited wax film?

Answer: In this paper, the experimental material microcrystalline wax is purchased and the material name is defined by the manufacturer (CAS NO: 63231-60-7).The crystallite size of microcrystalline wax 80#is less than 55 nm based on references. The research focus is not on the physical and chemical properties of microcrystalline wax materials, but on the printing and manufacturing process in this paper. The crystallite morphology of microcrystalline wax was not characterized and analyzed at present.

  1. Line 256: Physical parameter should be italic.

Answer: The Physical parameters has been italicized according to the reviewer's suggestion.

  1. Line 262: 0ms => 0 ms (Gap is needed. Check all)

Answer: The gap has been added between all numbers and unit symbols according to the reviewer's suggestion.

  1. Line 282: the copper clad plate was used as a deposition substrate. (Polarity of this surface and interaction with your wax?)

Answer: High-voltage-electric-field-driven jet deposition technology is a novel high resolution 3D printing, which is based on electrostatic induction and electrohy-drodynamic jetting behavior. The substrate material may be either a conductor or an insulator. The Polarity of wax surface and the copper clad plate surface are same under the high pulse voltage. The negative charge is distributed on the upper surface of the substrate, the positive charge is distributed on the lower surface of the substrate.

  1. Line 316: PET => Full name may be needed for introduction. Difference between the copper-clad plate and PET? All physical parameter should be italic.

Answer: The full name of PET (Polyethylene terephthalate) has been has been added to the article. The PET is a dielectric, but the copper clad plate is a conductor. The generated electric field is stronger under the same pulse voltage between the meniscus wax liquid surface and the copper clad plate. All physical parameters have been italicized according to the reviewer's suggestion.

  1. Table 2: Printing substrate: only PET? Or copper-clad plate together?

Answer: The Printing substrate is only PET film in Table 2.

  1. Line 406: Wax microstructure? Authors did not show any microstructure (e.g., nano/micro-scale morphology of a solidified wax film). Simply, authors may think that micro-scale deposit of wax solid on a substrate is a wax's microstructure. However, in Materials Science, when we say “microstructure, it has a different meaning.

Answer: The description of “wax microstructure” is really inaccurate, the wax microstructure has been revised into micro-scale wax structures.

We appreciate for Editors/Reviewers’ warm work earnestly, and hope that the correction will meet with approval. Once again, thank you very much for your comments and suggestions.

Sincerely yours,

Yanpu Chao, Hao Yi

[email protected]

[email protected]

Reviewer 2 Report

This is interesting work on micro-featured 3D printing using electrohydrodynamic phenomena. However, I recommend to make the following changes for accepting the manuscript for publication:

  1. You need to get the full manuscript screened by professional services or an English native speaker. For example, the title should start with "Experimental Analysis" instead of "Experminent Analysis"
  2. The Abstract, line 17: the abstract should not quote the pulse width variable, (Wt) because it is not needed to understand the idea and is distracting.
  3. Lines 37, 61, 91, etc.: the authors use multiple times the sentence "at home and abroad" that does not line up with a journal with an international audience (whose home? the authors'? why is important to make the distinction?). Furthermore, the text is unnecessary. The authors should remove those references from the text.
  4. Lines 37 to 45: the Introduction should include the micro-nano 3D printing technique called microplasma deposition. Relevant papers to quote include
    1. Y. Kornbluth et al,  Nano-Additively Manufactured Gold Thin Films with High Adhesion and Near-Bulk Electrical Resistivity via Jet-Assisted, Nanoparticle-Dominated, Room-Temperature Microsputtering, Additive Manufacturing, Vol. 36, 101679 (11pp), 2020. doi: 10.1016/j.addma.2020.101679
    2. A.M. Abdul-Wahed et al, Direct writing of metal film via sputtering of micromachined electrodes, J. Mater. Process. Tech. 262 (2018) 403–410, doi: 10.1016/j.jmatprotec.2018.06.028.

    3. T. Wang et al, Microplasma direct writing of a copper thin film in the atmospheric condition with a novel copper powder electrode, Plasma Process. Polym. 17 (2020), E2000034, doi: 10.1002/ppap.202000034. 

  5. Lines 58 to 68: the description of previous work on electrohydrodynamic printing should include key applications demonstrated by the technology. The references quoted in such text should include
    1. Y. Jiang et al, All electrospray printed perovskite solar cells,
      Nano Energy, Vol. 53, 2018, pp. 440-448, doi: 10.1016/j.nanoen.2018.08.062
    2. B. García-Farrera et al, Sub-100 nm Ceramic Piezoelectric Films via Electrohydrodynamic Deposition for Printed, High-Frequency Applications, ACS Applied Materials & Interfaces, Vol. 11, No. 32, pp. 29167 – 29176, August 2019. doi: 10.1021/acsami.9b09563
    3. A. P. Taylor et al, Electrospray-Printed Nanostructured Graphene Oxide Gas Sensors, Nanotechnology, Vol. 26, No. 50, 505301 (8pp), 2015. doi: 10.1088/0957-4484/26/50/505301 
  6. Line 70: the text "the printing process needs to ensure that the nozzle is conductive material" is plainly not true. The liquid can be used as filed enhancer to trigger the Taylor cones; the actual nozzle does not need to be electrically conductive, and, in many cases having a dielectric nozzle helps operate the device at a lower bias voltage.
  7. Figure 2: Please move all lables outside regions with color because they are very hard to read. Specifically, there are two labels on top of a gray table (deposition substrate, XY-axes motion platform) that should be relocated to white space.
  8. Section 4.2, the discussion should also comment on using other methods for controlling the wetting front in the nozzle, for example, coating the nozzle with a hydrophobic film. That is commonly used to avoid the liquid to drip on the sides of the emitter. Again, the nozzle does not need to be electrically conductive.
  9. Section 4.3, the discussion of pulsed vs continuous voltage should be extended to explain the idea of analog vs digital and how this affects the prints. When using pulses, the print is made of discretized into voxels, similarly to the way information is discretized and reduced to fall within a number of possibilities (digital), which makes it more robust as there are just so many possibilities; in contrast, when using a continuous voltage, the print is time controlled and the size of the voxels covers an infinite range of possibilities (analog). For controlling what will be printed, the best case is to have pulsed voltage and a very small voxels (grayscale).
  10. The authors agree that the printing is electrohydrodynamically induced; did the authors try to measure the current emitted? even if is a very small current (nA level), you should be able to measure it. It would help understand several things that are not clear from the results, for example, how the electrical relaxation time compares to the hydrodynamic time to form the droplet, deposit the droplet, etc. Measurements of the current would significantly improve the paper.
  11. Lines 281 to 289, 300 to 308 etc.: the authors need to be consistent throughout the paper: sometimes they quote the experimental values on both the text and in tables, sometimes they use one of them. Only one is needed, choose either one (refer to experimental conditions in Table X, or state the experimental conditions in the text, but do not do both).
  12. Lines 286 to 289: plot the droplet diameter versus pulse width. Is there a fit that describes the data? (linear? power law? semi-log? log-log?)
  13. Lines 286 to 289: what are the error bars of the measurements? Specifically, what is the uncertainty on the time of the pulse (Y ms+/- X ms) and the diameter (A um +/- B um)? How many measurements were done for each data point reported? All this needs to be reported.
  14. Lines 309 to 315: plot the droplet spacing versus speed of the stage. Is there a fit that describes the data? (linear? power law? semi-log? log-log?)
  15. Lines 309 to 315: what are the error bars of the measurements? Specifically, what is the uncertainty on the speed of the stage (Y m/s+/- X m/s) and the distance between droplets (A um +/- B um)? How many measurements were done for each data point reported? All this needs to be reported.
  16. Section 4.7: why the diameter of the nozzle was increased? you did some of your experiments with a nozzle diameter of 150 um, some others with a nozzle diameter of 200 um. Why? does this influence the results or is an unimportant detail? please explain.
  17. How large is the outer diameter of the nozzles at the tip? You declare the inner diameter, but the outer diameter is also needed to understand the kind of electric field acting on the meniscus.
  18. Given that you showed you can make continuous lines of droplets horizontally and vertically, can you make a more interesting structure that uses your printing method? for example, can you make a pyramid (or step pyramid), a dome, etc.? Something that shows true 3D, freeform capabilities. I imagine this should be straightforward to do if you program the stage. Showing such image(s) would significantly improve your paper and make the work very popular. 
  19. Section 5 (Conclusions): you should include only the highlights with high novelty of the work reported, not all you did. For example, having a high-speed monitoring system is something common (also, your optical system could use some more magnification). The key contribution of the paper is to demonstrate and characterize digital printing of wax using electrohydrodynamic phenomena that is pulsed.
  20. The paper should have a paragraph a the end (end of Section 4 (section 4.8) or at the end of Section 5) that explains how the technology you are developing can greatly increase the throughput via parallelization of the emitters. This is super important, it could increase two or more orders of magnitude the throughput without sacrificing the quality of the print, greatly extending the capabilities of the technology. For droplets/polymeric melts, key papers to report should include  
    1. W. Deng et al, Increase of electrospray throughput using multiplexed microfabricated sources for the scalable generation of monodisperse droplet, Aerosol Science 37 (2006) 696–714

Author Response

Dear Editors and Reviewers:

Thank you for your letter and for the reviewers’ comments concerning our manuscript entitled “Experimental Analysis of Wax Micro-Droplets 3D Printing Based on High-voltage-Electric-Field- Driven Jet Deposition Technology” (crystals-1595188). Those comments are very valuable for improving our paper. We have studied comments carefully and have made correction. Revised portion are marked in red in the paper. The revised details according to the reviewer's comments as follows:

>Reviewer #2:

Comments and Suggestions for Authors:

This is interesting work on micro-featured 3D printing using electrohydrodynamic phenomena. However, I recommend to make the following changes for accepting the manuscript for publication:

  1. You need to get the full manuscript screened by professional services or an English native speaker. For example, the title should start with "Experimental Analysis" instead of "Experminent Analysis"

Answer: According to your previous comments, we have carefully checked the whole manuscript for the English language, grammar, punctuation, spelling, and overall writing style. Meanwhile, this revised manuscript has undergone English language editing by MDPI. The text has been checked for correct use of grammar and common technical terms, and edited to a level suitable for reporting research in a scholarly journal (Please see the attached certificate of English editing). We hope that the revised manuscript will be sufficiently clear to express our ideas.

  1. The Abstract, line 17: the abstract should not quote the pulse width variable, (Wt) because it is not needed to understand the idea and is distracting.

Answer:The pulse width variable (Wt) has been deleted according to the reviewer's suggestion.

  1. Lines 37, 61, 91, etc.: the authors use multiple times the sentence "at home and abroad" that does not line up with a journal with an international audience (whose home? the authors'? why is important to make the distinction?). Furthermore, the text is unnecessary. The authors should remove those references from the text.

Answer: The expressions of “at home and abroad” have been revised according to the reviewer's suggestion.

  1. Lines 37 to 45: the Introduction should include the micro-nano 3D printing technique called microplasma deposition. Relevant papers to quote include.

(1)Y. Kornbluth et al, Nano-Additively Manufactured Gold Thin Films with High Adhesion and Near-Bulk Electrical Resistivity via Jet-Assisted, Nanoparticle-Dominated, Room-Temperature Microsputtering, Additive Manufacturing, Vol. 36, 101679 (11pp), 2020. doi: 10.1016/j.addma.2020.101679

(2)A.M. Abdul-Wahed et al, Direct writing of metal film via sputtering of micromachined electrodes, J. Mater. Process. Tech. 262 (2018) 403–410, doi: 10.1016/j.jmatprotec.2018.06.028.

(3)T. Wang et al, Microplasma direct writing of a copper thin film in the atmospheric condition with a novel copper powder electrode, Plasma Process. Polym. 17 (2020), E2000034, doi: 10.1002/ppap.202000034.

Answer: The above three papers of microplasma deposition have been cited in this paper according to the reviewer's suggestion.

  1. Lines 58 to 68: the description of previous work on electrohydrodynamic printing should include key applications demonstrated by the technology. The references quoted in such text should include.

(1)Y. Jiang et al, All electrospray printed perovskite solar cells, Nano Energy, Vol. 53, 2018, pp. 440-448, doi: 10.1016/j.nanoen.2018.08.062

(2)B. García-Farrera et al, Sub-100 nm Ceramic Piezoelectric Films via Electrohydrodynamic Deposition for Printed, High-Frequency Applications, ACS Applied Materials & Interfaces, Vol. 11, No. 32, pp. 29167 – 29176, August 2019. doi: 10.1021/acsami.9b09563

(3)A. P. Taylor et al, Electrospray-Printed Nanostructured Graphene Oxide Gas Sensors, Nanotechnology, Vol. 26, No. 50, 505301 (8pp), 2015. doi: 10.1088/0957-4484/26/50/505301

Answer: The above three papers of electrohydrodynamic printing have been cited in this paper according to the reviewer's suggestion.

  1. Line 70: the text "the printing process needs to ensure that the nozzle is conductive material" is plainly not true. The liquid can be used as filed enhancer to trigger the Taylor cones; the actual nozzle does not need to be electrically conductive, and, in many cases having a dielectric nozzle helps operate the device at a lower bias voltage.

Answer: This sentence of “the printing process needs to ensure that the nozzle is conductive material" has been deleted according to the reviewer's suggestion.

  1. Figure 2: Please move all lables outside regions with color because they are very hard to read. Specifically, there are two labels on top of a gray table (deposition substrate, XY-axes motion platform) that should be relocated to white space.

Answer: All labels have been moved outside regions with color in figure 2 according to the reviewer's suggestion.

  1. Section 4.2, the discussion should also comment on using other methods for controlling the wetting front in the nozzle, for example, coating the nozzle with a hydrophobic film. That is commonly used to avoid the liquid to drip on the sides of the emitter. Again, the nozzle does not need to be electrically conductive.

Answer: The new discussion of using other methods for controlling the wetting front in the nozzle has been added to the Section 4.2 according to the reviewer's suggestion.

  1. Section 4.3, the discussion of pulsed vs continuous voltage should be extended to explain the idea of analog vs digital and how this affects the prints. When using pulses, the print is made of discretized into voxels, similarly to the way information is discretized and reduced to fall within a number of possibilities (digital), which makes it more robust as there are just so many possibilities; in contrast, when using a continuous voltage, the print is time controlled and the size of the voxels covers an infinite range of possibilities (analog). For controlling what will be printed, the best case is to have pulsed voltage and a very small voxels (grayscale).

Answer: The new discussion of pulsed digital signal vs continuous voltage analog signal affects the printing has been added to the Section 4.3 according to the reviewer's suggestion.

  1. The authors agree that the printing is electrohydrodynamically induced; did the authors try to measure the current emitted? even if is a very small current (nA level), you should be able to measure it. It would help understand several things that are not clear from the results, for example, how the electrical relaxation time compares to the hydrodynamic time to form the droplet, deposit the droplet, etc. Measurements of the current would significantly improve the paper.

Answer: It is very necessary to measure the current emitted, I agree with the reviewer's suggestion. At present, the laboratory lacks relevant electric current measurement equipment, and the subsequent measurement research in this aspect will be carried out. 

  1. Lines 281 to 289, 300 to 308 etc.: the authors need to be consistent throughout the paper: sometimes they quote the experimental values on both the text and in tables, sometimes they use one of them. Only one is needed, choose either one (refer to experimental conditions in Table X, or state the experimental conditions in the text, but do not do both).

Answer: The problem of repeatedly quoting experimental values has been revised according to the reviewer's suggestion.

  1. Lines 286 to 289: plot the droplet diameter versus pulse width. Is there a fit that describes the data? (linear? power law? semi-log? log-log?)

Answer: The droplet diameter versus pulse width is an approximate linear relationship. A linear diagram has been drawn and added to the article.

  1. Lines 286 to 289: what are the error bars of the measurements? Specifically, what is the uncertainty on the time of the pulse (Y ms+/- X ms) and the diameter (A um +/- B um)? How many measurements were done for each data point reported? All this needs to be reported.

Answer: The analysis of parameter error bars and measurement errors have been added to the article according to the reviewer's suggestion.

  1. Lines 309 to 315: plot the droplet spacing versus speed of the stage. Is there a fit that describes the data? (linear? power law? semi-log? log-log?)

Answer: The droplet spacing versus speed of the deposition platform is an approximate linear relationship. A linear diagram has been drawn and added to the article.

  1. Lines 309 to 315: what are the error bars of the measurements? Specifically, what is the uncertainty on the speed of the stage (Y m/s+/- X m/s) and the distance between droplets (A um +/- B um)? How many measurements were done for each data point reported? All this needs to be reported.

Answer: The analysis of parameter error bars and measurement errors have been added to the article according to the reviewer's suggestion.

  1. Section 4.7: why the diameter of the nozzle was increased? you did some of your experiments with a nozzle diameter of 150 um, some others with a nozzle diameter of 200 um. Why? does this influence the results or is an unimportant detail? please explain.

Answer: It is an unimportant detail, two different nozzle diameters of 150 um and 200 um were deliberately selected to conduct experimental research.

  1. How large is the outer diameter of the nozzles at the tip? You declare the inner diameter, but the outer diameter is also needed to understand the kind of electric field acting on the meniscus.

Answer: The outer diameter of the nozzles has been added to the article. The outer diameter of the nozzle with inner diameter200 um is 300 um, the outer diameter of the nozzle with inner diameter 200 um is 350 um.

  1. Given that you showed you can make continuous lines of droplets horizontally and vertically, can you make a more interesting structure that uses your printing method? for example, can you make a pyramid (or step pyramid), a dome, etc.? Something that shows true 3D, freeform capabilities. I imagine this should be straightforward to do if you program the stage. Showing such image(s) would significantly improve your paper and make the work very popular.

Answer: The deposited line and micro-columns are indeed a little too simple and lack the typical morphology. The typical pyramid structure can be fabricated by programming to control the droplet deposition trajectory in experimental systems. But, currently, due to the COVID-19 outbreak, the laboratory has been closed and relevant experiments cannot be carried out temporarily. Relevant experiments will be carried out in the future. I hope the review experts can understand. Very thankful!

  1. Section 5 (Conclusions): you should include only the highlights with high novelty of the work reported, not all you did. For example, having a high-speed monitoring system is something common (also, your optical system could use some more magnification). The key contribution of the paper is to demonstrate and characterize digital printing of wax using electrohydrodynamic phenomena that is pulsed.

Answer: The Section 5 (Conclusions) has been revised according to the reviewer's suggestion.

  1. The paper should have a paragraph the end (end of Section 4 (section 4.8) or at the end of Section 5) that explains how the technology you are developing can greatly increase the throughput via parallelization of the emitters. This is super important, it could increase two or more orders of magnitude the throughput without sacrificing the quality of the print, greatly extending the capabilities of the technology. For droplets/polymeric melts, key papers to report should include.

(1)W. Deng et al, Increase of electrospray throughput using multiplexed microfabricated sources for the scalable generation of monodisperse droplet, Aerosol Science 37 (2006) 696–714

Answer: The related technology outlook about increasing the throughput via parallelization of the emitters have been added to the section 4.8 according to the reviewer's suggestion.

We appreciate for Editors/Reviewers’ warm work earnestly, and hope that the correction will meet with approval. Once again, thank you very much for your comments and suggestions.

Sincerely yours,

Yanpu Chao, Hao Yi

[email protected]

[email protected]

Reviewer 3 Report

This manuscript describes a high-voltage-electric-field-based micro 3D printing technique using crystalline wax as the printing materials. The printing process was comprehensively characterized, and the optimized ejection and deposition process parameters were identified. Overall, this manuscript is well written, and the reported system should be interesting to the community. There are several minor revisions recommended before publication.

1) A main issue of this paper is that there is no sufficient vendor information about the devices and materials used, like in section 3. There should be information about the product type, vendor name and country after each devices/materials introduced.

2) The font sizes in several figures are too small to read (Figure 2, 7, 9).

3) Page 1, line 37, “At home and abroad” could mean different things to different people, please correct the phrase.

4) Page 2, line 47, please add references for the limitations the author mentioned regarding to single material printing.

5) Page 4, line 159, please indicate the accuracy of the moving system and temperature control.

6) Page 6, line 216, the wrapping of the droplet is also due to its surface tension with the nozzle material. Please also clarify the direction of the negative pressure.

7) Page 9, table 2. Please clarify what does Plus/s mean for movement velocity.

8) in figure 8, please clarify the meaning of those dash lines

9) Page 10, figure 10, the author may want to zoom in to show the droplets.

10) Page 11, please add a scale bar in figure 11. Please clarify the material interaction mechanism of “fusion bond”.

11) Page 12, please elucidate how the points were chosen for diameter measurements

Author Response

Dear Editors and Reviewers:

Thank you for your letter and for the reviewers’ comments concerning our manuscript entitled “Experimental Analysis of Wax Micro-Droplets 3D Printing Based on High-voltage-Electric-Field- Driven Jet Deposition Technology” (crystals-1595188). Those comments are very valuable for improving our paper. We have studied comments carefully and have made correction. Revised portion are marked in red in the paper. The revised details according to the reviewer's comments as follows:

>Reviewer #3:

Comments and Suggestions for Authors:

This manuscript describes a high-voltage-electric-field-based micro 3D printing technique using crystalline wax as the printing materials. The printing process was comprehensively characterized, and the optimized ejection and deposition process parameters were identified. Overall, this manuscript is well written, and the reported system should be interesting to the community. There are several minor revisions recommended before publication.

  1. A main issue of this paper is that there is no sufficient vendor information about the devices and materials used, like in section 3. There should be information about the product type, vendor name and country after each devices/materials introduced.

Answer: The sufficient vendor information about the devices and materials have been added to the section 3 according to the reviewer's suggestion.

  1. The font sizes in several figures are too small to read (Figure 2, 7, 9).

Answer: The font size has been increased in figures 2, 7, 9 according to the reviewer's suggestion.

  1. Page 1, line 37, “At home and abroad” could mean different things to different people, please correct the phrase.

Answer: The expressions of “at home and abroad” have been revised according to the reviewer's suggestion.

  1. Page 2, line 47, please add references for the limitations the author mentioned regarding to single material printing.

Answer: The expressions of “single material printing” is inaccuracy, which has been revised according to the reviewer's suggestion.

  1. Page 4, line 159, please indicate the accuracy of the moving system and temperature control.

Answer: The accuracy of the moving system and temperature control have been added to the article according to the reviewer's suggestion.

  1. Page 6, line 216, the wrapping of the droplet is also due to its surface tension with the nozzle material. Please also clarify the direction of the negative pressure.

Answer: In this paper, the negative pressure balance gravity method is used to make the wax liquid form meniscus shape at the nozzle and maintain a stable meniscus shape. Negative pressure is formed in the crucible by negative pressure vacuum pump, and the direction of negative pressure is upward. 

  1. Page 9, table 2. Please clarify what does Plus/s mean for movement velocity.

Answer: The motors of XY axis moving platform adopts the control mode of "pulse + direction signal",   the meaning of Plus/s is the number of pulses sent to the motor by the controller in one second.

  1. Please clarify the meaning of those dash lines .Page 10, figure 10, the author may want to zoom in to show the droplets.

   Answer: The meaning of those dash lines is to guide the reader to the area of primary concern in figure 10.

  1. Page 11, please add a scale bar in figure 11. Please clarify the material interaction mechanism of “fusion bond”.

Answer: A scale bar has been added in in figure 11, the material interaction mechanism of “fusion bond” is that: after the surface of the previously deposited droplet is solidified, when the later deposited droplet is deposited on the solidified surface, remelting and bonding occur at the contact surface. 

  1. Page 12, please elucidate how the points were chosen for diameter measurements.

Answer: The choice of these points for diameter measurements is the location of the refusion region of two adjacent droplets in figure 15.

We appreciate for Editors/Reviewers’ warm work earnestly, and hope that the correction will meet with approval. Once again, thank you very much for your comments and suggestions.

Sincerely yours,

Yanpu Chao, Hao Yi

[email protected]

[email protected]

Round 2

Reviewer 1 Report

I recommend this revised version (Manuscript ID: crystals-1595188) should be published as it is because Authors well addressed what Reviewer requested. 

Reviewer 2 Report

I think the authors did a significant rework of their paper. My recommendation is to accept the paper in its current form.